# Three Perspectives on the Experience of Support for Family Caregivers in First Nations Communities

**DOI:** 10.3390/diseases11010047

**Published:** 2023-03-08

**Authors:** Amber Ward, Laurie Buffalo, Colleen McDonald, Tanya L’Heureux, Lesley Charles, Cheryl Pollard, Peter G Tian, Sharon Anderson, Jasneet Parmar

**Affiliations:** 1Faculty of Medicine, University of Victoria, Victoria, BC V6T 1Z3, Canada; 2Samson Cree Nation, Maskwacis, AB T0C 1N0, Canada; 3Enoch Cree Nation, Enoch, AB T7X 3Y3, Canada; 4Division of Care of the Elderly, Department of Family Medicine, University of Alberta, Edmonton, AB T6G 2T4, Canada; 5Faculty of Nursing, University of Regina, Regina, SK S4S 0A2, Canada

**Keywords:** Indigenous, family caregivers, qualitative, participatory action, First Nations

## Abstract

There is a dearth of research on how family caregivers are supported in First Nations. We interviewed family caregivers, health and community providers, and leaders in two Alberta First Nations Communities about their experiences of care and support for the family caregivers in their communities. We employed a qualitative, collaborative participatory action research methodology. We drew on Etuaptmumk, the Mi’kmaw understanding of being in the world is the gift of multiple perspectives. Participants in this research included family caregivers (*n* = 6), health and community providers (*n* = 14), and healthcare and community leaders (*n* = 6). The overarching caregiving theme is the “Hierarchy of challenge”. Six themes capture the challenges faced by family caregivers: (one) “Caregiving is a demanding job”: yet “No one in a sense is taking care of them”; (two) difficult navigation: “I am unable to access that”; (three) delayed assessments and treatment “And I don’t know how they’re being missed”; (four) disconnected health records: “It’s kind of on you to follow up”; (five) racism, “It’s treated differently”; and, (six) social determinants of health, “A lot of these factors have been developing for the longest time”. This study provides evidence that family caregivers’ need to care for and to maintain their own wellbeing is not top of mind in policy or programs in these First Nations communities. As we advocate for support for Canadian family caregivers, we need to ensure that Indigenous family caregivers are also recognized in policy and programs.

## 1. Introduction

One in four Canadians is a family caregiver (FCG) [1,2]. We define FCGs (carers and care partners) as any person, a family member, chosen family, friend, or neighbor who takes on a generally unpaid caring role providing emotional, physical, or practical support in response to mental or physical illness, disability, or frailty. In the Cree language, a caregiver/healer is Onatahwihwew and the business of a caregiver taking care of is Onakahtohkewin. While most (87%) adult children who are FCGs to their parents say giving care is rewarding [3], intensive care work, that is, caring for over 21 h weekly for people with severe impairments, dementia, depression, or anger or responsive behaviors can physically and psychologically drain FCGs [4,5,6]. American and Canadian scholars Taylor and Quesnel-Vallée theorize that the “structural burden of care”, which is becoming aware of, finding, negotiating for, and then managing siloed health and community social care services, adds significantly to FCGs burden [7]. Researchers indicate that indigenous FCGs’ caregiving experiences are more difficult than non-Aboriginal FCGs [8,9].

Delivery of aboriginal health services and continuing care is complex [10,11,12]. Minister Mark Miller’s 2020 report [13] and the 2018 report of the Standing Committee on Indigenous and Northern Affairs [14] noted that the “complicated and ambiguous framework” of overlapping responsibilities and current policies shared between federal and provincial governments, different ministries and departments, First Nations leadership, organizations, communities, and third-party service providers make it challenging for aboriginal people to access the services they need. Services also vary considerably across governmental levels and policy jurisdictions [13]. To date, the federal government has played a lead role in the delivery of services to First Nations living on reserves that would otherwise be delivered by provinces or territories [13]. The provinces and territories deliver services for First Nations living off First Nations. In some areas of provincial jurisdiction, Indigenous Services Canada has adopted alternative service delivery models involving bilateral or tripartite agreements, that specify distinct coordination of funding and service delivery [13]. Indigenous ways of being, knowing, and understanding are not necessarily incorporated in the design, development, or delivery of the services [13].

The complex First Nations health and continuing care delivery systems are also challenging for the healthcare providers who work with Aboriginal people and their families [14,15]. Some of the factors preventing health providers from meeting First Nations and Inuit peoples’ needs within the home care program included recruitment and retention of nurses, personal care providers, and other human resources; lack of training; understaffing resulting in inefficient care delivery; communication gaps with physicians, hospitals, and regional health authorities; and the challenges communicating with clients and families [15]. Indigenous Services Canada also identified increased demands from greater numbers of people with complex care needs, multiple chronic conditions, mental health, and addictions [15]. Although not voluminous, there is some literature on the experiences of Canadian indigenous caregivers [8,16], however there is a dearth of literature from FCGs’ and providers’ perspectives.

The aim of this paper is to report the experience of First Nations FCGs, health and community providers, and leaders of support and care for FCGs in First Nations communities.

### 1.1. Working with First Nations Communities

We began by meeting representatives from two Alberta First Nations communities to understand what might be useful for them to understand about FCGs. The representatives from both communities spoke about the complexities Aboriginal FCGs face getting services and the difficulty their health providers face trying to meet FCGs’ needs. They were very concerned about the differences between what First Nations FCGs living on and off the First Nation could access.

They also stressed that we needed to take a holistic, aboriginal social determinants of health approach to understand how caregiving links to larger levels of influence, such as Indigenous culture, policies (colonialism, housing, water, lack of funding for health care facilities, and community health/home care), finances (poverty, costs, and employment) services (transportation and service delivery organization) and practices. Beginning with the ceremony was important as was ensuring that we respect the space First Nations people share with us.

### 1.2. Context

People from two Treaty Six Cree Nations participated in this research. The Samson Cree Nation, (Cree: Nîpisîhkopâhk) is the largest of four band governments in Maskwacis in central Alberta. By the 2021 census, 3252 people live in the First Nation. The Enoch Cree Nation (Cree: Maskêkosihk) controls two reserves. The Enoch Cree Nation 135 is west of the City of Edmonton and 135A is south of the Town of Barrhead. The 2021 census population was 1825. Members of both Nations are of Cree ancestry and primarily speak the Plains Cree dialect of the Cree language group.

## 2. Materials and Methods

We employed a qualitative, collaborative, and participatory action research methodology. We drew on Etuaptmumk, the Mi’kmaw understanding of being in the world is the gift of multiple perspectives [17,18]. The Etuaptmumk view is that indigenous and nonindigenous views can be seen as complementary with researchers from both viewpoints considered as allies [19]. This approach seeks to recenter indigenous knowledge and concerns within knowledge [20,21]. Western knowledge is not rejected, rather it is seen as partial, a colonists’ record that needs to be balanced and resituated within indigenous histories and worldviews [21]. Throughout this research, we reflected on the unequal power relations that have historically dominated research and tried to address the unequal power relations by doing this project with aboriginal people, in ways that empower and benefit the people and the aboriginal communities participating. We recognize that there are over 600 different First Nations cultures and the Cree culture in the First Nations communities in our study is different from the Mi’kmaw culture; thus guided by community advisors, we followed the cultural protocols within each community. After receiving ethics approval from the University of Alberta Health Research Ethics Board, we began with a pipe ceremony to introduce the project to the community and seek community participation. Throughout this research, the aboriginal people in these communities were central in guiding the design, data collection, reviewing the findings, and developing recommendations.

### 2.1. Participatory Action Research

This research is the beginning of participatory action research (PAR). PAR involves researchers and participants working together to (1) understand the situation, the problems, and the strengths and (2) change it for the better. Typically, it is an iterative cycle of research, action, and reflection that seeks to raise participants’ awareness of their situation and works with them to take action. We are just beginning to work together on this research. Our goal was to build trusting relationships.

### 2.2. Participant Recruitment

This was a convenience, snowball sample. Community representatives advised community members about the research study. Community members interested in participating could contact the research coordinator or research assistant (1) by phone or email to arrange for an interview in the place and time of their choosing or (2) they could choose from posted dates when the research assistant would be in the community to conduct interviews. Participants were asked to tell others in their networks about the interviews. In these small communities where everyone knows everyone, several participants were particularly concerned about privacy.

### 2.3. Data Collection

Guided by our community advisors, people were invited to share their stories in individual interviews. Community advisors, a First Nations registered nurse with training and experience conducting qualitative interviews, and the research team designed an interview guide to encourage people to tell their stories of being family caregiving or being a health or community care provider caring for family caregivers or a health or community leader. It began with a general question about caregiving in the First Nation and then about their role, experiences, and what they thought might make caregiving or supporting caregivers easier (see Appendix A: interview guide). A First Nations registered nurse with training and experience conducting qualitative interviews completed all the interviews in addition to qualitative interview training which followed Olson’s [22] qualitative interview methodology. Aligned with qualitative participatory research methods, the research coordinator or the principal investigator, and the research assistant reviewed the interviews and discussed what might be explored in the next interviews. Interviews were conducted on ZOOM, by telephone, or in person, based on each participant’s preference. She followed community COVID-19 and World Health Organization protocols to protect participants in in-person interviews. Participants received $30 honoraria. Interviews lasted 35 min to 75 min.

### 2.4. Analysis

Interviews were transcribed verbatim and then checked for accuracy and cleaned of any identifying information. They were imported into NVIVO for data management. We used Braun and Clarke’s [23] thematic data analysis methods. Thematic analysis is a flexible qualitative method used to explore the different perspectives held by research participants; it highlights the similarities and divergences in their viewpoints and generates thematic insights [23]. Braun and Clarke’s [23] analysis proceeds in six stages. To become familiar with the data and to generate first impressions of meaning (stage one), two members of the research team independently read participants’ responses and made notes of their impressions in memos in NVivo. In stage two, members of the research team worked separately to inductively generate initial open codes. In stage three, team members worked together to generate categories. Patterns within the open codes were identified and codes with similar attributes and meanings were grouped. The categories were then refined into preliminary themes (stage four) using critical questions such as “what is happening here” and “what is being said here” and “why”? At stage four, the community advisory and research teams discussed how the knowledge might apply in these First Nations communities, influence health providers’ work with FCGs, and how this knowledge might influence practices and policies. We then reread the transcripts to name and confirm the final themes (stage five). We generated a report (stage six) and shared it with the communities.

## 3. Results

Participants in this research included FCGs (*n* = 6), health and community providers (*n* = 14), and healthcare and community leaders (*n* = 6). All FCGs identified as First Nations and providers and leaders as First Nations, Cree, Canadian, Caucasian, Filipino, and Black. All were over 21 and two were over 65 years of age. Over half of the providers currently were or had been FCGs. The roles they identified with included: family caregiver, dental therapist, community health worker, director of health, medical transportation coordinator, manager, registered nurse, licensed practical nurse, program officer, chief, manager, health policy, health services advisor, health manager, lawyer, advanced care paramedic, home care nurse, outreach coordinator, and doctor. In what follows, first, we situate the research with the participants’ descriptions of FCGs, then move to the themes.

The overarching caregiving theme is the “Hierarchy of challenge” in which aboriginal FCGs, health providers, and healthcare leadership were coping with siloed and complex community, healthcare, policy, and funding systems. Six themes capture the challenges faced by FCGs: (one) “Caregiving is a demanding job”: “No one in a sense is taking care of them”; (two) difficult navigation: “I am unable to access that”; (three) delayed assessments and treatment “And I don’t know how they’re being missed”; (four) disconnected health records: “It’s kind of on you to follow up”; (five) racism, “It’s treated differently”; and (six) social determinants of health, “A lot of these factors have been developing for the longest time”. These challenges also impact providers’ ability to care for FCGs. We place FCGs’, providers’, and leaders’ exemplar quotes side by side in Table 1 to show the convergence.

### 3.1. Situating Family Caregiving and Who Needs Care

We began the interviews by asking participants about “caregivers” (“Who were the caregivers in the community?”) and care receivers, (“Who needs the most care?”) and to tell us about caregiving and receiving in ‘this community’.

#### Caregiving: “It’s Part of Our Nature”

Participants described caregiving as part of the Cree culture: “We just do it because it’s part of our nature”. They explained how it was their role to care for people needing care when called on. Both caregiving and needing care were seen as valued roles given by the creator, the caregiver’s, and the care receiver’s purposes at this time in their life. FCGs, health providers, and leaders all explained that the entire family are caregivers. The family included the immediate family, such as grandparents, parents, children and siblings, maternal and paternal aunts and uncles, cousins and other kin, as well as all people in the community. “We are all intertwined”. Building on the broad view of caregiving as part of the Cree cultural identity and aboriginal kin of all ages as caregivers, participants also included the employed health and community providers as caregivers.

### 3.2. Overarching Theme: “Hierarchy of Challenge”

Despite providing a holistic view of care and caregivers, as the interviews proceeded, participants told stories of a siloed approach to care and siloed supports at all levels of influence. FCGs, health providers, and leaders all talked about the difficulty of dealing with many different levels of government including the Federal Indigenous Services Canada (ISC) and First Nations and Inuit Health Branch (FNIHB), Provincial Alberta Health, Alberta Seniors and Housing, Alberta Persons with Developmental Disabilities, Alberta Family and Community Support Services, the Health Centre in their First Nations community, Alberta Health Services hospitals, homecare, and long-term care, primary care and medical specialists off reserve, First Nations leadership and administration, as well as the programs and services offered in the community (e.g., home care, elders’ supports, housing, ambulance, and medical transportation). One community leader described coping with numerous, complex systems as the “hierarchy of challenge” with ISC and FNIHB at the top, First Nations leaders “almost at the top”, and vulnerable FCGs “at the bottom”,


*The systems themselves are very much built on a hierarchy, a Western type of colonialist mentality. So, for whatever reason, and this has been ongoing for a number of years now, where elected leaders are almost at the top, and then the most vulnerable people who actually put us into office seem to be at the bottom. And then in-between there’s all these programs and services like this hierarchy of like challenge.*

*[Leader]*


#### 3.2.1. Theme 1: “Caregiving Is a Demanding Job” Yet, “No One in a Sense Is Taking Care of Them”

Participants acknowledged that being a First Nations FCG was demanding work. Even though the family is important, these participants portrayed one person as the primary caregiver. FCGs spoke to being on their own as they learned about the illness or condition and as they tried to access services and support. Health and community providers and leaders concurred that FCGs were taking on complex care primarily without healthcare or community support. FCGs, providers, and leaders all talked about FCGs needing knowledge about illness and education, about the conditions the person they were caring for, as well as supports to care for and to maintain their own wellbeing. A leader pointed out that while most patients had FCGs accompanying them, there was no specific department or program with responsibility for FCGs, and there was no care specifically for FCGs. The leader hoped that training health providers would “trickle-down” to caregivers. See Table 2: Exemplar Quotes by Theme: Family Caregivers, Health and Social Care Providers, and Leaders

#### 3.2.2. Theme 2: Difficult Navigation, “I Am Unable to Access That”

All participants spoke about the difficulty that both FCGs and health and community providers had navigating health and community systems. In participants’ descriptions, FCGs were responsible for learning about and managing the care receiver’s health condition, caring for, and navigating the health and community care systems. FCGs also had to advocate for and coordinate care from each health and social care provider, program, and department. All participants reported that the healthcare and home care of First Nations offered a greater range of services. They provided examples of outpatient medical services, rehabilitation, home care, and others that were more accessible and comprehensive if people moved off the First Nation. Many participants talked about how limiting First Nation’s home care to “banker’s hours”, weekdays from 9–5, was not meeting FCGs’ needs.

Caregivers, providers, and leaders all talked about advocating for better services for FCGs. Notably, the FCGs participating in this research said that they were advocating for others who were afraid to or did not have the skills to advocate for themselves. Providers were aware of the system’s shortcomings so suggested navigators, a team approach, and systemic changes based on the FCG’s and patient’s needs would improve navigation. A leader connected the difficulty navigating specifically to siloed federal, provincial, and First Nation responsibility for services and supports.

#### 3.2.3. Theme 3: Delayed Assessments and Treatment, “and I Don’t Know How They’re Being Missed”

Every caregiver told stories of delayed diagnosis, assessments, and treatments. Caregivers of children with a variety of conditions—cerebral palsy, autism, or a cleft larynx–spoke about delayed diagnoses. The health provider who provided palliative care for her mother and then her sister talked about a “gap” in which Aboriginal people “were not being tracked” and “getting missed” from hospitals to services in the First Nation. Health providers connected delayed assessments and treatments to caregivers’ difficulty navigating the system. After discussing similar concerns as FCGs and providers, a leader related healthcare delays to the systemic racism built into funding mechanisms, which he contended left aboriginal people having to justify why they needed services and expecting to be treated differently.

#### 3.2.4. Theme 4: Disconnected Health Records, “It’s Kind of on You to Follow Up”

After two caregivers attributed some of the difficulty, they were having to access services to disconnected, inaccessible healthcare records, we explored participants’ experiences with health records and how that affected patients and caregivers. Health and community providers and leaders confirmed that different levels of government were using a variety of different health records systems. It appeared to depend on health providers to check health records from other providers who were using and coordinate continuity of care.

#### 3.2.5. Theme 5: Racism, “It’s Treated Differently”

All the caregivers related racism to how they as caregivers were treated. One caregiver grappled to find the words to describe being treated differently by health providers at the local hospital in comparison to urban hospitals, while other caregivers spoke directly to racism in being “accused” of not caring appropriately or in the way they were treated. One caregiver, several health providers, and leaders linked racist outcomes, such as the FCGs’ difficulty accessing support in a timely manner and being turned away at dental offices, to racism. The senior leader recommended talking openly about racism in the healthcare system.

#### 3.2.6. Theme 6: Social Determinants of Health, “A Lot of These Factors Have Been Developing for the Longest Time”

FCGs, health providers, and leaders all referred to upstream social determinants of health such as poverty, housing, safe water, lack of trust in the system, and experiences of racism that made caregiving more difficult. Although a health provider noted that there was no place in the policy for health transfer policies for FCGs, she observed that not supporting FCGs was not an isolated problem. She estimated the “intermingling” of issues and “convoluted” processes that prevent care for FCGs likely affected 95% of the over 600 Canadian First Nations. One leader ventured that access to programs and services was not equitable for people in poverty and that there were gaps in all programs that needed to be addressed.

## 4. Discussion

In these First Nations Communities, participants characterized Onakahtohkewin, the business of a caregiver taking care of someone, as a chronic unaddressed challenge that takes a backseat to acute challenges. For the most part, FCGs are not being taken care of, rather they are at the bottom of a “hierarchy of challenges”. As a health leader noted, “there is no department for family caregivers”. FCGs are not recognized in the convoluted multilevel layers of existing siloed policies and programs that dominate indigenous people’s lives. The Truth and Reconciliation Commission of Canada’s Calls to Action [10] are beginning to bring about change in cultural recognition and policies, however, as these participants pointed out, family caregiving is not a pressing issue.

Indigenous FCGs, like all Canadian FCGs, are essential to support the social connections, dignity, and wellbeing of the people they care for [7,24,25,26,27]. In traditional Cree culture, family caregiving is valued for the purpose that the role gives to the caregivers’ lives and for the value of caregiving to the individual and the community as a whole. FCGs’ care work is also crucial to sustain the formal health and social care system [28,29,30]. Canadian FCGs provide 90% of the care to people living in private community homes [31] and assist with 15 to 30% of the care in supportive living and long-term care [32,33]. Many of the participants in this study suggested that First Nations FCGs are likely providing more care, as they expected to provide care without the support available to nonindigenous Canadian caregivers [16]. Indigenous FCGs urgently need integrated policy, health, and social care systems to support them to care for and maintain their health.

In addition to providing care, these First Nations FCGs were struggling with a much greater structural burden of care [7]. The structural burden of care is the time spent identifying the services and supports needed, looking for and accessing the services, and then coordinating services. At the point of care, FCGs and the people they care for were dependent on health providers coordinating disconnected federal, provincial, and community health records and relaying that information to them so they become knowledgeable about the health condition with which they are dealing and then finding, negotiating for, and managing siloed health and community social care services. Health providers stressed that many times, coordination at the point of care was lacking. They too were stymied by the complicated records systems and policies between different government levels. Racine and colleagues’ [16] review of the experience of indigenous caregiving for dementia related to the burden of geographic isolation, lack of funding for healthcare facilities, and legacies of colonial policies and trauma. We agree that these factors also increase the indigenous structural burden of care. Most participants noted that the supports such as home care, rehabilitation, or respite were more comprehensive and accessible through provincial healthcare than on the First Nation. All levels of governments need to ensure that indigenous FCGs are not disadvantaged because they live in a First Nation or Metis Settlement.

Navigation was extremely difficult for these First Nations FCGs as well as the health and community providers who were trying to support them. All participants provided examples of FCGs trying to access support, however, being put on long waiting lists, told that they had to apply to another level of government for coverage, shuffled out without treatment, or missed completely. The difficulty navigating contributed to delays in diagnosis, treatment, rehabilitation, and timely access to support. Timely rehabilitation or treatment for children with cerebral palsy, autism, or other conditions is essential for them to thrive. Jordan’s Principle holds that First Nations children should not be denied access to public services while governments fight over who should pay [34,35,36]. Adults also need timely preventive care, diagnosis, and treatment. Providers suggested a navigator might reduce some of the difficulties both they and the FCGs faced trying to access health services.

Participants all discussed racism that ranged from being treated differently to overt discrimination in which aboriginal people were denied services. They tied unaddressed social determinants of health like poverty, inadequate housing, unsafe water, and health issues to colonial policies and systemic racism. Systemic racism is associated with inconsistent and low-quality care for indigenous people [16]. Racism is a potent psychosocial stressor that combines with other factors to increase poverty, reduce access to employment, reduce mental and physical health, and increase premature mortality. [37,38] Indigenous people need access to culturally informed and equitable health care, which might also help to address indigenous health inequities [38,39,40]. Bruno [39] suggests that experiential learning is likely superior to cultural awareness training. Moreover, as there are more than 650 recognized First Nations with diverse cultures, languages, and histories he recommends a community-based approach.

### 4.1. Strengths and Limitations

Triangulating the views of FCGs, health and community providers, and healthcare and community leaders is a strength of this study. We were surprised by the congruence in their views. The interviews were conducted by a First Nations health provider who developed a rapport with the participants to produce very rich interviews. These two communities are both Cree and have similar traditions. While they are rural, they are not remote and relatively advantaged compared to other Alberta First Nations. We expect that some of the themes would be different in other communities, particularly those that are remote, fly-in communities. Based on this and other research, we think that the FCGs’ needs receive less attention than other acute needs in most indigenous communities. FCGs are referred to as “Invisible” and “the backbone of the health system” [28,41]. We need to make First Nations and Metis FCG’s work visible too.

The Standing Committee on Indigenous and Northern Affairs report on the challenges of delivering continuing care in First Nations communities [14] stressed that little research had been done on continuing healthcare services on First Nations. This study was exploratory and there is much more that needs to be done in these communities and other remote First Nations communities. A strength of this research is that it revealed the difficulty FCGs, providers, and leaders have in dealing with the complex array of government policies and records systems.

### 4.2. Implications

Although we did not assess these aboriginal FCGs’ health and wellbeing, worldwide, FCGs are at increased risk of deteriorating health [42,43] and financial insecurity [44] which makes the FCG role itself a public health issue and social determinant of health [42,45]. Both Public Health England [42] and the American National Caregiving Alliance [45] point to a greater proportion of FCGs than noncarers with chronic conditions and had difficulty taking care of their own health. Framing family caregiving from the public health perspective positions issues related FCGs’ work and role as a lifecourse population health concern. We know that if the caregiver is healthy, the quality of care and life of the person receiving care will be substantially improved. We did a report to the community on 24 January 2023 and will work with the communities to finalize and then report on the recommendations.

## 5. Conclusions

This study provides evidence that FCGs’ needs to care for and maintain their own wellbeing are not top of mind in policy or programs in these First Nations communities. As we advocate for support for Canadian FCGs, we need to ensure that indigenous FCGs are also recognized in policy and programs. There should be a department that explicitly ensures that FCGs are considered.

## Figures and Tables

**Table 1 diseases-11-00047-t001:** Demographics.

Ages	
20 years and under	
21–25	1
26–34	5
35–44	5
45–54	7
55–64	5
Over 65	2
**Gender**	
Woman	21
Man	5
**Ethnicity**	
First Nations	15
Cree	2
Caucasian	5
Black	1
Filipino	1
Prefer not to answer·	2

**Table 2 diseases-11-00047-t002:** Exemplar Quotes by Theme: Family Caregivers, Health and Social Care Providers, and Leaders.

Family Caregivers	Health and Community Providers	Leader
Situating family caregiving and Who needs care.		
Caregiving, “It’s part of our nature”.		
I am a Cree Nation member from a musical tribe. I am 56 years old and have always been a caregiver. I think I was born into it. [Caregiver]	I know there’s one family member. She lost her dad and mom and her siblings. She’s the only living person in their family with all of the grandkids now …. So she had to take on that role. How it works is all of my sister’s and brother’s kids are my kids. And she is the only one left technically, in the creative life and family values, that is her role. [Health Provider]	And it wasn’t really a pride thing. It was almost like you signed up for this. This is your husband. This is the cards the creator has given you. You will do this. [Leader and Caregiver]
“We are all intertwined”		
I am a caregiver to my son [Name]. I’m one of his caregivers. We all play a role. His immediate family. So, my husband does. And so do my two older children [ages young adults]. We all love [Name]. So we all share in that. And we also get my two sisters. They live nearby. [Caregiver]	I think for us it really is, you know, the unity of my family. It’s not just my immediate family that loves and cares for [Name of sibling], but it’s all of our family. My extended cousins are always looking out for him. …And I witnessed him with second and third cousins and extended family, and they’re the exact same way. [Health Provider and Caregiver]	When healthcare talks about person-centered care we kind of miss the boat, because in our First Nations cultures, we’re so close and our family units are so expansive, we’re all intertwined. … So patient-centered care is important, but because of the interconnectivity of our families, the family is just as important. It’s not just happening to their loved one, it’s happening to you and the whole family at the same time. [Leader]
Theme 1: Caregiving is a demanding job, “No one in a sense is taking care of them”		
There’s a lot of lot of things to consider when you’re taking care of your dependent adult child in your home. That’s what we as caregivers go through. If anything, what I can say for myself as a caregiver is the lack of rest, the lack of sleep. People say, “Oh, you got to take care of yourself. You’ve got to come first [Caregiver Name]”. And there’s most times I turn away and I drive away crying and I just say to myself, “If only you people knew how hard it was to get free” time for myself. So the biggest factors in my life are the lack of sleep and the finances because of the cost of living is making it hard to get through the month. Because I also have my grandchildren to take care of as well. [Caregiver]	I guess the caregivers in this community face a lot of struggles, one maybe being a lack of resources, even racism. And then another thing is they tend to do everything alone. And I think a lot of times that comes from, I guess, us having to have a person who is disabled, whether there were physical, emotional, psychological disabilities. We raised them without having to resort to anybody’s assistance. The caregivers in this community today really need assistance. Some of them can’t get it here [Name of Health Centre] for example, if they’re raising a disabled child. [Community Provider]	So caregiving is a full-time job. And it’s a demanding job. It’s a job, you know. And if there is no funding in place for training on reserve to promote aging in place, you know, the responsibility is left with family caregivers. Nobody else comes to their aid. [Leader] Because they’re caregivers, their tendency in my view, is to overlook their own personal health because of the tasks they take on in urgent situations. They take on events or activities or incidents that can be very traumatic and are taken on by the caregiver with no one, in a sense, taking care of them. So there is a need I see to support caregivers every which way possible. [Leader]
Primarily without support		
[Professional/Paid] caregivers in this community are really few and far between. Having been struck with RA (Rheumatoid Arthritis), in [year] I found there was nobody that was available to even tell me what the disease was about and what to expect and the whole bit. And when my husband, started experiencing his health problems, that went to a different level. There’s virtually no services for people in this community. He was on heart meds, you know, he had two heart attacks, barely survived. I had to pay for his medication because he’s Metis. We, my family, were really traumatized. [Caregiver & community provider]	I don’t think they get anything [any training or support] unless they go off reserve. And I really don’t understand what kind of programs that we would provide except for maybe how to lift a person up off a wheelchair. I remember working in a nursing home when I was going to college and they taught us all that how to make beds, lift patients and feed them and everything else. Basically, here, I think it’s just a daily life experience. When we’re living with somebody who needs our assistance on a daily basis, we get up, we feed them, we hug them, we love them. It’s just a regular routine. [Caregiver & Health Provider]	Well off the top of my head there is not a specific program that looks at caregiver education or caregiver support. As far as I know, right now we’re looking at what we can do to support the nurses. And we’re hoping for that trickle-down effect if we can get the nurses supported and have the education and the knowledge that would trickle-down to the client and the caregivers, because the caregivers come in with the clients most of the time. So that would be the first barrier there’s no specific program that I know of for caregivers. [Healthcare Leader]
Theme 2: Difficult navigation: “I am unable to access that”		
We’re still where we were in 1985, when my mother passed away. No nurses. I had to change the bandages on my mother’s breast, and I was only 17 years old, and I had to work the at pool hall. So, I’d work during the day and then I’d go home and clean her bandages and cook for her. And nobody taught me what to cook or how to manage her pain. And now again with my sister, who just recently passed away… I had to reach out to them and say, “Hey, listen, my sister has cancer and she’s dying. And I know it’s going to take her life, but I need some help here”. And they’re like, “Oh, she’s not even on the radar”. [Caregiver and Community Provider] They offered a program to bring my child to Edmonton to get assessed, put in a special school, but I would have to live in a city because it would be a daytime program, and with my children, I’m unable to move. I’m unable to access that. And they were trying to help her get past her anxiety, [other identifying conditions]. [Caregiver]	I think just being told what’s going on. Sometimes the family, they take their families to the hospital, and they don’t know what’s going on. Or maybe the doctors or nurses explained, but maybe they may need to involve somebody like a physician’s support in the hospital to explain things well. Because a lot of patients come to me like, “I don’t even know the plan for my child”. I’m like, “I thought you were in the hospital. Did they explain?” “No, they didn’t explain to me anything”. [Health Provider] I think there’s lack of [services] for autism, not labeling it, but stating that they’re in that Autism Spectrum area. Learning how to address it and how to teach them to be sociable in these areas. And I know one of our colleagues has a child in that area, but he had the opportunity to go get training in the States. I don’t think there’s any training here. I know there’s another band member who has [child] on the autism spectrum, and they had to move to be near a place that has some type of training …. So there is no training. [Health Provider]	We are restricted to a lot of where we can move money and how we can help our community members. Yeah, the government says that they give us how many millions of dollars, but then they separated, they divide up, and then the majority goes to ISC. Or. And then we have follow these rules and regulations like when we give home care we can hire someone to come into your house and clean, but it can’t be your family member. But that’s all that people trust is their family members. They don’t want someone to come into their home and look around because of all the trauma that they had experienced through residential school. We’re only allowed one person. And how can we give adequate services to our elders, to our vulnerable, to our disabled with only one person. It takes two people to bath them or lift them. The rules restrict how we deliver our services to our community members and there’s just there’s not enough funding. [Healthcare Leader]
Always advocating for better services		
And what really angers me about this and the point that I’m trying to make is my husband and I, we can argue for services. We can advocate. But there are lots of families out there who can’t. They don’t have the advocacy skills. They don’t have the [pause] I don’t want to say courage because I think all families who go through this, who have handicapped people in their families, they always have to have courage. But there’s a lot of families who are afraid. They’re afraid of institutions. They’re afraid of the government. They’re afraid of authority, of anybody in power. So a lot of people don’t speak out. I shouldn’t need to be a lawyer in order to get somebody’s attention, in order to get my issues addressed. I just think about all of the families who don’t have advocacy skills. And I have relatives who have had to put their kids in care because they didn’t have the resources and they didn’t have the supports to keep their loved ones at home. And they’ve had to put their children in care. And so, as you know very well, that’s what a lot of families end up doing. They either have to move off the reserve or they put their kids in care. [Caregiver]	I see kids in wheelchairs and with unique transportation needs, and I have families with four or five children where more than one child has complex needs. And I can see that like families are struggling and often there’s a history of trauma and all sorts of other things as well. And it’s heartbreaking to watch a family who, based on social determinants of health has such an uphill battle and then is dealing with super complex medical issues on top of it all. So, they need help with all of those things, I mean, travel in and out of the city, accommodation, a navigator who would be able to help them, just orient them to the system where they go for what, where these buildings are where they’re supposed to go for specialist appointments and how to get around the city. Some system level coordination, so care would be coordinated so families wouldn’t necessarily have to have these desperate kinds of appointments all spread out and have to make lots of trips in and out for separate appointments. But to be able to have somebody who you could call on when people needed care, that the health care system is just not that equipped to provide, language services. Oh, gosh, I could go on for a long time. [Health Provider]	With caregiving comes, I think, a very specific expert knowledge for that particular individual. And that is not really always taken, I think, properly. And the caregiver has not just the role of the caregiving, but they’re serving an advocacy role as well because they’re advocating for their home to be fixed or they have to leave their homes to go move off reserve so that they can get better service within Alberta Health Services programs. But they want to stay home, they want to be in community because when you’re in community, you have access to elders, you have access to family, you have access to ceremony. And, you know, and it’s just nice knowing that you’re in your homeland. But there’s not enough to help them, even something as simple as meals. We don’t have a Meals on Wheels service here. So things like that that aren’t even the radar. Everybody’s looking at the immediate crisis needs. And caregivers are not a crisis and when you’re focusing on immediate crisis needs, you can’t get to proactive planning. [Leader]
Theme 3: Delayed assessments and treatment, “And I don’t know how they’re being missed		
There’s a gap where there’s people that have these illnesses and they’re not being tracked. They’re being missed. And I don’t know how they’re being missed, especially if they come from a huge facility, say the [Name] of cancer facility or whatever hospital. Yeah they are getting missed. [Caregiver] I mean, I always want to find out things. I was always reading up on pregnancy. And anyway, I figured out that he had cerebral palsy fairly early on. But the doctors would never confirm that. It wasn’t until he was about five or six that they finally confirmed it. But anyway, on a scale of 1 to 5 he’s a 4. This is how the doctor explained it to me on a scale of 1 to 5, with one being normal and five being catastrophic. [Name] is a four. [Caregiver] The baby that I traditionally adopted when she was born, she was really sickly when I got her. She was just kind of given to me without any information. I knew that at four months old she was in and out of the hospital three or four times with pneumonia. Right away we knew that she was unable to lay flat because she would start aspirating. So, we had to put her at a 45 degree angle. And I took her back to the Pediatrician and told her what was going on. The Pediatrician was really good, but I’d say a month later, she got really sick again. And we got taken to [hospital]. I was treated like I did something wrong. They even accused me of just putting a bottle in her mouth and leaving her to suck the bottle on her own with something propped up against it. And here it was, I was an Indian. There was a lot of racist comments against me and that wasn’t the case. I took care of her like I had carried her right through [pregnancy]. Two years later, we finally found out it was a cleft larynx, so she was aspirating her food and drink for two years. Why was that missed? [Caregiver and Community Provider]	So we get a lot of requests for things like FASD assessments or children with special needs, where I have concerns about developmental assessments and developmental pediatrics, and there’s a lot of delay and trouble getting those assessments in a timely manner. Mostly x-ray and lab and whatever is pretty straightforward. It’s mostly diagnostics, like where you’re diagnosing somebody with a syndrome where you need the advice of a specialist. It’s not something that I can, like do an x-ray and say, “This is the problem”. It’s somebody who has that specialized knowledge. I can tell that it’s not normal, but it’s outside the scope of family medicine and then it’s like trying to connect them to the appropriate resources. [Health Provider] And my experience is that they’re not that well connected. They come to me and they have a lot of different issues that I am not equipped to handle. I can tell them, for this you have to go here or for this let me call this person and we need to bring them into it. So I can kind of be a little bit of a navigator and an advocate, but at the same time, I don’t have the expertise to deal with all the issues myself. Sometimes I feel like I’m a bit of a quarterback, but I’m not necessarily like the whole team, you know? If I’m the whole team, we got a problem. We need a bigger team. [Health Provider]	And, you know, and then we consider the Indian Act, right? And then we consider the systemic racism and discrimination that is built into those funding mechanisms, along with the societal systemic bias and racism that when our people try to go access service, be it preventive, be it emergency, be it maintenance of health, our people as Indigenous Peoples, number one, are always expecting to be treated differently or to be treated like they’re being a bother for wanting to make sure that their person that they’re caring for or themselves or their child or whoever, that when they’re accessing that service, they feel like you’re going to have to defend something. You feel like you’re going to have a whole list of really great things to defend. And it’s almost like you’re validating why you’re there, seeking the support, and seeking the assistance. And you can’t just have one reason. It’s like we have to have a whole bunch. [Senior Leader]
Theme 4 Disconnected health records, “It’s kind of on you to follow up”		
We do not have access to our own health records, nor is it in our possession at our own health centers. Everybody else has our records. We need those back. [Caregiver] There’s the gap in our records, in services, and in coverage. It is impossible to keep track. This is why Jordan’s Principle came about and became an issue because the province took the view that these kids are a Federal Health responsibility. And Federal Health, the Feds said, well, this the provincial health responsibility. So everywhere we turned, we were told, “You got to go talk to somebody else. You’ve got to go see someone else”. So we felt like ping pong balls. You know, we were always told, “Go ask that person, I can’t help you”. So, all of the government officials nobody ever said, “I’m going to help you. And I’m going to find these answers for you”. That never happened. Not once with anybody. [Caregiver]	Right. So, there is a provincial communication records system. So, a lot of people on reserve are not able to access it, but also there wasn’t any way that physicians off reserve can access Indigenous records. So, it’s like you’re dealing with double edged sword. [Health Provider] Right now, we have access to Netcare, but there’s not really a formal communication system between Alberta Health Services and the band or the health centers. If you sent the patient out, it’s kind of on you to follow up and check Netcare to see what happened to them when they went out. No one will really call you unless it’s quite urgent, like, “I’m sending this patient back. We amputated their leg. You probably need to see them tomorrow”. They don’t really let us know. They’re quite busy. Again, they’re also short staffed. Limited staffing is probably the big thing, but we don’t really have much communication. [Health Provider]	No, our records aren’t connected. So even Alberta Health Services, we have different health care management. So, we have Phantom, we have Meditech, we have Netcare. All of those things are actually going into Connect Care and then there will be one system. But for FNIHB, we have something called CHIP or C-DOM for a communicable disease management. So FNIHB is a bit connected to system, but not really. So CHIP, we can extract information from Netcare. So let’s say if one of our clients goes into an Alberta Health Services Clinic to get immunized, the nurse enters it into Meditech. Well now. Before it didn’t go to Netcare, but now it’s going to Netcare because of COVID. We need those records. And now in CHIP, we have that option of extracting Netcare’s immunization record and merging it into our CHIPS records. So, we do have a bit of that connection, but we don’t have any connection to Phantom, Meditech, or even ConnectCare at this point. [Healthcare Leader]
Theme 5 Racism, “It’s being treated differently”		
I don’t know if mistreated would be the right word. It’s being treated differently. I would have to say yes, going back to when I was going to school, treated differently. My perspective of the health system is that Aboriginal people are treated differently. My son was diagnosed with [Name of illness] in [rural town]. We went to the U of A, for a lot of treatment. From there, we went to Calgary Children’s Hospital, where we went underwent treatment again. And so, there was an apparent treatment difference between the metro hospitals, the metro health professionals and going just for follow up checkups for bloodwork in [local rural town name]. Oh, my gosh. It was like night and day. I could not believe the difference between the local hospital and those big hospitals. Even my son knew it at 15. He’s seen it and he realized, we are not treated well at [rural town]. This is what we’re dealing with here locally. [Caregiver]	The caregivers really struggle. There’s racism out there and some of them feel like they have to move off reserve just to get the service from the province. I think we need to change those views. We need to make it better for taking care of our own people. [Health Provider] More needs to be implemented in terms of racism and discrimination and what it looks like, what is taught and the fact that it actually has an effect on our people’s health and wellbeing. That racism restricts where they can go for care. For example, First Nations are covered with treaty status for dental and some offices refuse clients with that insurance. [Health Provider] The Indian Act still has the mentally incompetent Indians where, God forbid, if his caregivers passed away, then he becomes a ward of the state. Where’s the kinship in that? That’s not part of our community, that’s more levels of control. Like we need to have something that’s culturally relevant to our people to protect our most vulnerable people. And we need to protect our relatives. [Health Provider]	Yes, we had residential schools. Certainly, it affected many of us. A few not so much. It has a domino effect where there are issues related to parenting and illness, and knowing how the larger society views you as a population. Of course, it’s going to have a negative impact. When that whole negative stereotypical view of those individuals who ran those residential schools came out, the racism got more evident. You know they thought of us as savages. They thought of that we’re no good. [Leader] There is racism in the health institutions. There is racism in the denial of adequate health services to our people because of their race. It’s easily hidden just by someone who says “I’m not willing to serve that patient” and it’s easily denied. We need to come face to face with that issue. We need to be able to talk openly, candidly, honestly about the racism that exists in the health care system as far as Indigenous Peoples are concerned, in particular women and children. That would be foundational. [Senior Leader]
Theme 6: Social determinants of health, “A lot of these factors have been developing for the longest time”.		
And programs and services need to recognize that a lot of these factors have been developing for the longest time, and nobody’s addressed how they impact today’s health and wellbeing of individuals in a community. [Caregiver] The ramp that was supposed to be done in a professional way. It was not approved. It’s not up to code, but there’s no funds to fix it up. Housing always says there’s no funds. No funds, grandchildren sleep downstairs, and I have mold in the basement. I need a new water well, no funds. We all bathe in unclean water because there’s no funds to do water wells. [Caregiver]	It really tough working on the First Nations Reserve. I find that a lot of trauma from residential school, trauma from growing up in poverty, and less than desirable conditions and that affects every age group out here. So, I would say everybody out here needs a caregiver or at least someone to talk to. [Health Provider] I think of the caregivers in my role, really addressing families at bedside. But when I consider leadership in terms of chief and council’s and government’s role especially with the top-down approach and the staggering intermingling of all these issues, I guess everything is kind of convoluted process. Even talking in terms of the racist policies, that are enforcing what I do and what happens in communities. And where I’m going to try to go with this, is that for support for caregivers is like this across the country. Like there are 600 plus First Nations across Canada that are probably the same boat, probably more likely 95% are in the same boat because we fall under federal jurisdiction and there is no place in policy, especially with the health transfer for family caregivers. [Health Provider]	So it concerns me, knowing the abject poverty that many of our people live with, and whether or not their access to the service or the program that’s supposed to be available is actually accessible. And because of that reserve funding structure, many of our services hold banker hours, they only work, say 8 to 4 or 9 to 5 or something. And we all know that life happens outside of those hours. So, you know, we end up having a window greater than two-thirds of the day where many of our caregivers or somebody who’s living with a disability don’t have access to service or program. [Leader] Unless there’s investment made in every single piece of a healthy community housing, clean water, education, and health, their social determinants of health, it’s the continuum of life, as a man, as a girl, as a boy, as a teenager, as a young mother, as a young father. And despite the myriad of programs and services you’ll find gaps in every area because it’s not comprehensive in nature or is not viewed as being important by the larger society.... Right. And that investment not only made within the programs and services, it has to be made within the capacity development of the people. [Leader]

## Data Availability

This intellectual property belongs to the Nations. The Nations have the autonomy to decide how, with whom, and when this information is shared. Please email sdanders@ualberta.ca to inquire about access.

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
