# Peer review of "Three Perspectives on the Experience of Support for Family Caregivers in First Nations Communities"

_diseases, 2023, doi:10.3390/diseases11010047_

Round 1

Reviewer 1 Report

Thank you for giving me the opportunity to review the article. The author conducted a qualitative study focusing on the family caregivers in First Nations Communities. The topic is socially important, but there are several methodological problems in the manuscript. Therefore, the reviewer thought that the manuscript should be revised before further considerations. I listed my comments below.

Comments:

1.      Is the “Sampson Cree Nation” typo of the “Samson Cree Nation?”

Methods:

2.      The authors should explain about the First Nations Communities focused in this study. The potential readers cannot about the populational and the cultural backgrounds and contexts.

3.      How did the authors recruit the participants? They should describe the details.

4.      The authors should describe the details of the semi-structured interview conducted in this study.

5.      The authors should provide the guide used for the interview as a supplementary material.

6.      The authors should describe the process to maintain the quality of the interview (e.g., training for interviewer).

Results:

7.      The authors should provide the backgrounds of the participants as the Table 1. Currently, they only provided very limited information.

8.      The authors should describe about the characteristics (time and others) of the interviews.

Discussion:

9.      The authors should discuss about the limitations in more details.

Author Response

Reviewer # 1
Thank you very much for these suggestions. We have been very concerned about ensuring confidentiality in these close-knit communities. We likely went overboard. This study was meant to be exploratory, to develop relationships and trust. Over the last two years as we have done Provincial presentations, the Metis and First Nations Caregivers told us that they had a difficult time accessing services. However, it takes time to develop relationships and get permission from leadership. I have worked with Amber Ward since 2007 and she is hands down the best interviewer I have worked with. If the interview guide seems light, it is because she is so good about getting participants to expand on their story. After a family caregiver said, “We need our records back” Amber asked her about all about the records she was interested and why. She followed up on that in subsequent interviews.
1. The p in Samson is a typo. Thank you
Methods
2. We have provided some context on lines 95 to 105 on the two First Nations Communities in this study.
3. We added section 2.1 Recruitment and described recruitment methods in lines 131 to 139.
4 We described construction of the semi-structured interview Lines 142- 148
5 The interview guide is included as a supplementary file.
6. The First Nations Nurse and the Research Coordinator have both been trained by Dr. Karin Olson, who wrote the textbook cited on qualitative interviewing. The research assistant and coordinator both listened to the interviews and reviewed the transcripts
then discussed what needed to be improved and where more curiosity would have produced a richer interview. Lines 150-154.
Results
7. These are very small communities where everyone knows everyone. We were reluctant to provide more information about participants in case we violated our promise of confidentiality. We have added as much information as we think we can. Lines 186-190 We also added Table 1: Demographics
8. We added the time spent in the interviews. Lines 159-160
Discussion
9. We added more depth and breadth to the limitations. Lines 385 to 399 and also an implications section Lines 400 to 410

Thank you so much for your suggestions. I think they have strengthened the paper. 

Reviewer 2 Report

Thank you for giving me the opportunity to review this interesting work. The paper was well written and I have few minor comments that I hope authors could address them prior to further evaluation.

1. How did the authors conclude with the available sample participants to conclude the results? Was saturation reached with the available sample size?

2. Under 2.2, authors need to mention that interviews were "transcribed verbatim."

3. How was participatory action research methodology conducted? It would be good to have detailed descriptions as a separate sub-section.

4. I believe that the methodology could be facilitated with a brief flow chart on the conduct, analysis approaches of the study.

5. The discussion could be further strengthen by elaborating the implications of the study for practice and suggestions for improvement through relevant public health policies.  

Author Response

Reviewer 2
Thank you for your excellent advice. We were concerned with privacy in these two communities because everyone knows everyone. We promised participants we would ensure they could not be identified in the reporting. We may have been too cautious. I think we have addressed all your concerns.
1. We have added that this study was exploratory, and population was a convenience snowball sample. Lines 132-133. These two First Nations are quite similar, many of them said they had cousins on the other Nation. Even though we have a full range of health providers and leaders, we cannot say we reached saturation the way saturation is typically used in qualitative research. We were hearing very similar stories and we were not coding new codes however we don’t want to over-reach.
2. We have added verbatim to transcribed.
3. Thank you. We have added Section 2.1 on Participatory research. Lines 132-139. This is the first step in the participatory process. We worked together to identify what research and how to proceed. We developed the interview guide collaboratively. We did a report to community in January and we are working to finalize another paper on recommendations.
4. This is a great idea, I would like to save this for the paper on recommendations that we have in progress.  
5. Thank you, for the suggestion to add implications. The First Nations and Inuit Health Branch’s responsibility is Public Health yet they “don’t have a department for family caregivers.” Both the UK and US have recently positioned caregiving as a public health/ social determinants of health issues. Lines 400 to 410. 

Thank you so much! I really appreciate the time that it takes to complete these reviews. 

Round 2

Reviewer 1 Report

Thank you for giving me the opportunity to review the article. The author revised the manuscript according to the comments. Therefore, the reviewer thought that the manuscript can be accepted for publication for the journal Diseases.